# Consumer's Attitude towards Display Google Ads

**Mohammad Al Khasawneh [1], Abdel-Aziz Ahmad Sharabati [2,\*] , Shafig Al-Haddad [1] , Rania Al-Daher [1], Sarah Hammouri [1] and Sima Shaqman [1]**

1    King Talal School of Business Technology, Princess Sumaya University for Technology, Amman 11941, Jordan
2    Business Faculty, Middle East University, Amman 11831, Jordan
\*    Correspondence: apharmaarts@gmail.com or asharabati@meu.edu.jo

**Abstract:** The context of Display Google ads and its components has significant importance to previous studies. However, the full understanding of the variables that influence both Display Google ads avoidance and intention to click has not been thoroughly acknowledged. Thus, this study aims to outline an entire understanding of the different variables that lead Display Google ads to be avoided or clicked on. A detailed review of previous studies has been completed to illustrate a thorough image of Display Google ads. Accordingly, this study developed a theoretical model combining four variables (Display Google ads' Prior Experience, Originality, Relevance, and Credibility) that lead to affecting Display Google ads' Avoidance and Intention to Click, with one mediator (Consumer's Attitude). A quantitative methodology has been employed, in which an online survey has been used to collect data, which were collected from 358 respondents, then coded against AMOS. The data analysis results show that three independent variables positively impact the intention to click; however, credibility has the highest value, then relevance and originality, consequently., while Display Google ads prior experience had no impact on the intention to click. Finally, the research concluded different practical and theoretical implications, and future potential research, and limitations.

**Keywords:** display Google ads; online marketing; advertisements avoidance; intention to click; consumers attitude

## 1. Introduction

Background: Due to the fast development of technologies and the Internet, the number of online users is increasing dramatically [1], especially the use of social media via smartphones among young people is growing fast [2]. Currently, all countries around the world are concerned about both media literacy and digital literacy, and they have developed their own policies to expand the knowledge of both media and digital literacy [3]. At the same time, social media tools are increasingly leading to changing marketing strategies and approaches [4]. The use of social media is becoming more important for all organizations because it can be used as a strategic marketing tool [5] to improve the relationship with customers [6] by providing customers with updated information and services [1,7] and enhancing customer engagement [8]. Marketing social networking sites have been developed by organizations to build stronger relationships with online users [9]. Google advertising platforms were first introduced back in the 2000s, which acted as a medium between online users and organizations [10]. Currently, advertisers are using Google Ads as an advertising platform [11]. Google Ads uses pull marketing to attract customers toward brands, engagement, and buying [12]. Digital marketing changes marketing tools, therefore, organizations use online marketing to attract customers [12]. Social media customer perception leads to customer engagement and defines how to respond to ads [12]. Advertisements through social media tools affect consumer attitudes [13]. The Google search engine market has 87.66% of the market share while Bing and Yahoo have only 12.33%, so organizations are increasingly using Google Ads [14]. Google and Facebook Ads are the wide tools to

affect brand awareness and customers' buying decisions [15]. Currently, Meta's Facebook and Google's free services are the most widely used by organizations to target customers through social media advertising [16]. Google ads affect customers' attention and behavior through cognitive responses [17]. Organizations seek to enhance inbound marketing efforts to satisfy customers with the needed information [18].

Research questions: Limited studies acknowledged examining the new category of Google Ads, which is the Display Google Ads. The current research aims to explain this new category by studying the variables that influence the effectiveness of this new advertising technique. Therefore, this research seeks to examine consumers' attitudes toward Display Google ads. Hence, the developed statement was chosen using the following objective of the study "to investigate the influence of Display Google Ads and all related factors that impact the consumer's attitude and customer's intention to click on Display Google Ads". To test this statement, a set of variables were examined that have a vital influence on the consumer's attitude towards Display Google ads. Accordingly, the main objective of the current study is to answer the following:

Q1: To what extent do consumers' related factors and Display Google Ads related factors influence consumers' attitudes towards Display Google Ads?

Q2: To what extent does the consumer's attitude influence the intention to click?

Q3: What are the most influential variables that affect consumers' attitudes?

The findings of the present research put forward useful theoretical and practical implications. Initially, the current research will fill the gap that was revealed in the literature review concerning limited acquaintances of the Display Google ads concept. However, the current study will add to other existing studies regarding this gap by constructing an unprecedented theoretical model that will be beneficial for other researchers. Thereafter, the assistance of the research will be helpful and advantageous in that it demonstrates the relation between Display Google ads factors that were not discussed significantly in previous studies and how those factors have the most influence on consumer attitudes.

### 1.1. Literature Review

Google Ads are the platform used by advertising agencies to provide an advertisement [11]. Google's search engine market has 87.66% of the market share while Bing and Yahoo have only 12.33%; thus, Google increases the organizations' opportunities to better contact customers [14]. Currently, Meta's Facebook and Google's free services are the most widely used by organizations to target customers through social media advertising [16].

Online advertisements increase recognition and purchase intentions if the user was exposed to the advertisement repeatedly [19]. Many previous studies focused on the consumer's attitude toward Sponsored Search Advertising (SSA) [20,21]. Other studies examined consumer behavior in the context of SSA [22–25]. Some studies concluded that brand image and key phrases are the most important variables that influence the consumer's behavior toward SSA [23], along with technology, innovativeness, and trustworthiness [24]. Another research declared that positive consumer attitudes toward SSA lead to a powerful image value in the long run [25]. Other studies investigated variables affecting the intentions to click regarding SSA and the chosen keywords that enhance click-through rates [26–29].

Google Ads uses keywords to define the effectiveness of ad promotion and product sales [11]. Google Ads affect customer perceptions of products and services with either a good image or a bad image [12]. The online ad develops customer perception to respond through a click, like, and share the content, and to buy the service and/or product [12]. There is a significant relationship between the Google ad and consumers' attitudes [13]. Google and Facebook Ads affect brand awareness and customers' buying decisions [15]. The success factors of using Google ads as a marketing tool include relevance, content, information, and experiences, which affect customers' behavior [14]. The privacy issue is the main concern of users when using social media [16]. Personal experience and confidence perform the main role to trust the advertising platform of Google Ads search engine

marketing [18]. Finally, there are limited studies that focused on consumers' attitudes towards Display Google ads in terms of intention to click [10,30].

Based on the previous discussion, it is clear that SSA took a major role in the literature. Additionally, other studies concentrated on the click-through rates as the only dependent variable; however, limited studies focused on examining the consumer's attitude towards Display Google ads regarding the intention to click. After a thorough examination of the relevant literature, it was concluded that further investigation of the factors influencing the consumer's attitudes toward Display Google ads, in particular, is very limited and is highly needed, as such justifying the aim of this study to fill this gap.

*1.2. Hypotheses Development and Research Model*

1.2.1. Display Google Ads Creativity: Ad Relevance

As for advertisers, examining ad relevance helps to improve the efficiency of a search page [31]. Online advertisements that are not placed relevantly on the Internet could affect the user negatively [32]. Creative and relative advertisements seek to get more attention and lead to more positive behaviors [33,34]. The more the advertisement was of interest to the user the fewer chances it will drive ad avoidance [35]. Google ad relevance is very important and its impact on consumers' attitudes [33]. Some previous studies showed the negative effect of annoyance and intrusiveness in terms of online advertisements, which can drive down behaviors toward the brand and lead to advertisement avoidance [36]. Advertisers select keywords related to their ad content [11]. For successful online marketing, the keywords should focus on relevance [14]. The most relevant information performs the main role to trust the advertising platform of Google Ads search engine marketing [18].

It is noted that limited studies explored the influence of ad relevance in terms of consumers' attitudes towards Display Google ads. Due to this, it is suggested that:

**H1.** *Display Google ads relevance has a significant positive influence on consumer attitude towards Display Google ads.*

**H2.** *Display Google ads relevance has a significant negative influence on Display Google ads avoidance.*

1.2.2. Display Google Ads Creativity: Ad Originality

Ad originality is a non-traditional form of advertising in which it develops positive intentions rather than using traditional strategies. Originality was analyzed in the context of consumer perception [33]. The importance of ad originality is that consumers pay more attention to a unique online advertisement; the unique placement of an ad on a certain platform can dramatically increase the effect on the consumer's perception of the advertising message positively [33]. Google Ads uses pull marketing to attract customers toward brands, engagement, and buying [12]. Social media ads increase consumer liking, awareness, and action, which have a strong correlation with consumers' attitudes. In this study, we have [13]. The keyword focus on originality, relevance, information, and content which affect customers' behavior [14]. Massage fit, frame, and focus affect customers' attention and behavior [17]. Real-time Google Ads create new ad data [37]. Original and relevant information increase trust [18].

Finally, in this research, Display Google ads originality can be defined as how well the advertisement can be distinguished by being unique and unparalleled. Accordingly, the following hypothesis is completed:

**H3.** *Display Google ads originality has a significant positive influence on consumer attitude towards Display Google ads.*

**H4.** *Display Google ads originality has a significant negative influence on Display Google ads avoidance.*

### 1.2.3. Display Google Ads Credibility

Credibility is the extent to which a consumer has perceived information and background with the advertisement [38]. Ad credibility was defined as the credible content of advertisements that made the ad more persuasive and convincing [39]. The trustworthiness and attractiveness of an ad can highly impact a consumer's attention [40], for example, a consumer's first impression of an ad can extremely influence its credibility of it, thus making it an important variable to be examined [39]. Positive thinking about the ad's credibility enhances customers' attitudes toward the ad [41]. Google ads create a positive image of products and/or services, and increase customer engagement and sales [12]. Trust in social media tools affects consumer attitudes [13], brand awareness, and customers' buying decisions [15]. Confidence and personal experience are important for customers' trust [18]. In the current study, credibility is described as the extent to which Display Google ads are believable to the user. It is noted that limited studies examined the effect of ads' credibility in terms of consumer attitude towards Display Google ads. Consequently, it is posited that:

**H5.** *Display Google ads credibility has a significant positive influence on consumer attitude towards Display Google ads.*

### 1.2.4. Prior Experience with Display Google Ads

Prior experience, within the context of consumer behavior, was defined as the information gained from experiences within SSA that act as a vital response in addition to being a predictor for future engagement [38]. Prior experience is a key factor that affects the consumer attitude in terms of intention to click [22]. Several studies found that positive prior experience could increase the impact on SSA credibility and attention to SSA [38]. A negative prior experience enhances the likelihood of an ad being avoided [42]. Prior experience is important as it depends on who sends and who receives the message [43], as the information that the receiver of the message has can perform a crucial role in the acceptance of the message [44]. Customers' prior experience and trust affect customers' attitudes and behaviors [18]. For this research, prior experience is defined as a set of previous knowledge and experiences that form consumer attitudes toward the perceived Display Google ads. It is noted that limited studies examined the effect of prior experience in terms of consumer attitude towards Display Google ads. For this reason, the following hypotheses are postulated:

**H6.** *Prior experience with Display Google ads has a significant positive influence on consumer attitude towards Display Google ads.*

**H7.** *Prior experience with Display Google ads has a significant negative influence on Display Google ads avoidance.*

### 1.2.5. Consumer Attitude towards Display Google ads

Advertisements must be a combination of three elements, which are credibility, trustworthiness, realism, and originality to create a positive consumer attitude [45]. Unfamiliar or irrelevant brands in an ad affect customer attitudes [46]. Analyzing consumers' attitudes is an important factor when examining the consumer's response toward an advertising message [24]. Google ads' keywords content determines customers' attitudes and the effectiveness of the ad promotion [11]. Google ads privacy influences Google ad Settings, which affects attitudes toward the websites [47]. Pull marketing affects customers' intention to use social media. Customer perceptions about products and services are either a good image or a bad image affect customers' attitudes toward Google Ads [12]. Advertisements through social media tools increase consumer liking, awareness, and action affecting consumer attitudes [13]. Google and Facebook ads affect brand awareness and customers' buying decisions [15]. Google ads improve customer contact and attitudes, which affect customers' behavior [14]. Organizations target customers through social media advertising to increase information and attitudes toward products and services [16]. Positive customer

attitudes toward Google ads increase trust and customer satisfaction [18]. For the current research, consumers' attitudes towards Display Google ads are the indicators of feelings and reactions of the consumer based on the factors that were listed earlier towards the advertisement. In light of this discussion, limited studies focused on consumer attitudes resulting in the intention to click on Display Google ads. Accordingly, the following hypothesis has been completed:

**H8.** *Consumer attitude towards Display Google ads has a significant positive influence on intention to click on Display Google ads.*

The suggested hypotheses are directed to test how Display Google ads' originality, relevance, credibility, and prior experience affect customer's attitudes toward Display Google ads, then Intention to click.

### 1.3. Study Model

Based on the previous discussion, the study model demonstrates the key factors that influence consumer attitudes regarding their intentions to click on Display Google ads. It is assumed that several elements impact consumers' attitudes toward SSA, such as ads prior experience, credibility, and relevance with brands and websites [38]. By trying to apply these factors to study consumers' attitudes toward Display Google ads, the model for the current research was further extended by including variables from different previous studies [33,38,48]. However, another study analyzed the leverage of creativity, originality, and relevance in the context of online advertisements [33]. Nonetheless, a study declared that consumer attitudes could give a rise to ad avoidance in online advertisements [48]. Finally, this study includes both intentions to click and avoidance to click. Accordingly, the following model shown in Figure 1 has been developed:

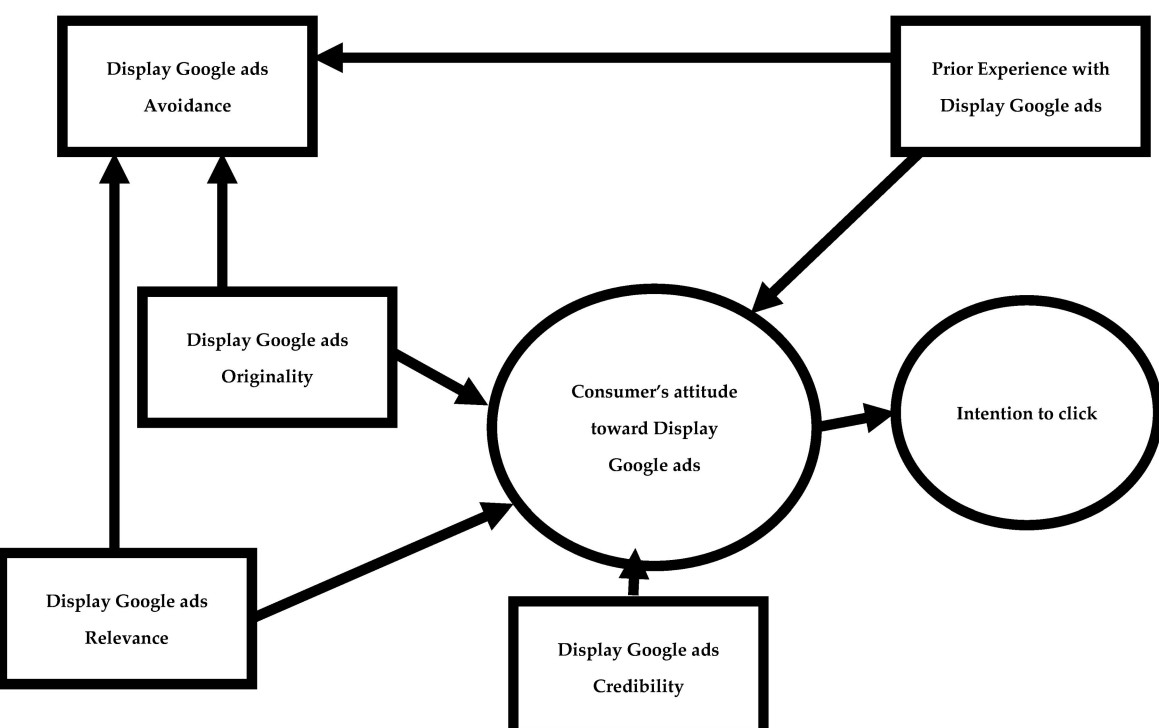

**Figure 1.** Initial model of variables affecting consumer's attitude.

## 2. Materials and Methods

### 2.1. Quantitative Data Collection

The current study uses quantitative cross-sectional research [49]. Quantitative research is described as a sort of empirical approach to social problems, which is a standard tool with

a set of planned classifications of various answers and predicts future outcomes [50]. Quantitative methods are useful when analyzing the demographics of the sample, consumption patterns, and consumer attitudes, and intentions. It includes collecting data, which could be useful for statistics and outcomes [51]. The tools of data collection can be either surveys or questionnaires resulting in numeric answers [50]. Survey research is widely used by researchers to collect data because it is simple and less time-consuming [42], and has low costs and minimal errors [52]. Online surveys are capable of including diverse questions for instance, multiple-choice questions, open-ended questions, and scales [53]. Therefore, the current research uses the online survey method to collect answers to questions, where the results tend to be more accurate and useful [54].

### 2.2. Survey Design

The online survey was developed to examine a sample of the population and their understanding of the main variables (Google ads' originality, relevance, avoidance, credibility, and prior experience). The survey for this research was created and designed through understandable and specific questions. The survey is comprised of two sections: the first section is made up of demographic dimensions, which consist of age, gender, and educational level. The second part of the survey is made up of the following variables: Prior experience with Display Google Ads [22], credibility of Display Google ads [22], Display Google ads relevance [22], Display Google ads originality [55], Display Google ads avoidance [56], consumer attitudes toward Display Google ads [22], and lastly, intention to click [22,57]. A Likert scale has been adapted to rate the agreement level from strongly agree to strongly disagree [58]. The survey consists of 27 closed-ended questions.

### 2.3. Data Collection Procedures and Sample Characteristics

The current research aims to examine the variables that impact the consumer's attitude towards Display Google ads. Online questionnaires were developed based on the previous literature. Non-probability sampling has been used to actualize this study. Data were collected from 432 diverse individuals who use the internet. A total of 74 surveys were excluded as answers were they do not notice Display Google ads, resulting in 358 surveys being valid for statistical analysis. Finally, structured equation modeling via AMOS 22 was used to analyze reliability testing, model validation, and model convergent validity. Respondents' demographic characteristics, age, gender, and educational level, are shown in Table 1.

**Table 1.** Sample Characteristics.

| Gender | Frequency | Percent | Valid Percent | Cumulative Percent |
|---|---|---|---|---|
| Male | 162 | 45.3 | 45.3 | 45.3 |
| Female | 196 | 54.7 | 54.7 | 100.0 |
| **Total** | **432** | **100.0** | **100.0** | |
| **Age** | **Frequency** | **Percent** | **Valid Percent** | **Cumulative Percent** |
| 16 years or under | 2 | 0.6 | 0.6 | 0.6 |
| 17–24 | 280 | 78.2 | 78.2 | 78.8 |
| 25–34 | 44 | 12.3 | 12.3 | 91.1 |
| 35–44 | 6 | 1.7 | 1.7 | 92.7 |
| 44–54 | 8 | 2.2 | 2.2 | 95.0 |
| 55 years or above | 18 | 5.0 | 5.0 | 100.0 |
| **Total** | **432** | **100.0** | **100.0** | |
| **Educational Level** | **Frequency** | **Percent** | **Valid Percent** | **Cumulative Percent** |
| High School Student | 34 | 9.5 | 9.5 | 9.5 |
| High School Diploma | 32 | 8.9 | 8.9 | 18.4 |
| Bachelor's Degree | 264 | 73.7 | 73.7 | 92.2 |
| Master's Degree | 20 | 5.6 | 5.6 | 97.8 |
| PhD | 8 | 2.2 | 2.2 | 100.0 |
| **Total** | **432** | **100.0** | **100.0** | |

The demographical data in Table 1 show that the amount of females are 196 and the amount of males are 162, this indicates that females participated more than males. The majority of respondents are aged between 17 and 24 years, with a percentage of 78.2%. As for educational level, the majority of the study sample were in their Bachelor's degree, and their number was 264.

## 3. Results

### 3.1. Reliability Test and Validation of Model

3.1.1. Exploratory of Factorial Analysis (EFA)

Exploratory Factorial Analysis (EFA) is a statistical analysis commonly used to test variance linear correlation with latent dimensions [59]. The criteria that specify the rule of thumb when conducting an EFA are factor loadings of more than 0.50, and cross-loadings of less than 0.30 [60,61]. Factor loadings lower than 0.30 should be excluded [62,63]. A factor loading above 0.40 is considered stable [63,64].

When applying criteria that specifies the rule of thumb when conducting an EFA: Table 2 shows that factor loadings more than 0.50, and cross-loadings less than 0.30, 7 items (questions) were deleted. Four items related to Avoidance and four items related to prior experience.

**Table 2.** Items Rotated Component Matrix.

|   |   |   | 1 | 2 | 3 | 4 | 5 | 6 | 7 |
|---|---|---|---|---|---|---|---|---|---|
| 1 | Credibility | C1 | 0.741 | | | | | | |
| | | C2 | 0.795 | | | | | | |
| | | C3 | 0.814 | | | | | | |
| | | C4 | 0.921 | | | | | | |
| 2 | Relevance | R1 | | 0.674 | | | | | |
| | | R2 | | 0.774 | | | | | |
| | | R3 | | 0.769 | | | | | |
| | | R4 | | 0.851 | | | | | |
| 3 | Originality | O1 | | | 0.877 | | | | |
| | | O2 | | | 0.881 | | | | |
| | | O3 | | | 0.863 | | | | |
| | | O4 | | | 0.901 | | | | |
| | | O5 | | | 0.931 | | | | |
| 4 | Avoidance | A1 | | | | 0.578 | | | |
| | | A2 | | | | 0.914 | | | |
| | | A4 | | | | 0.796 | | | |
| | | A6 | | | | 0.687 | | | |
| | | A7 | | | | 0.814 | | | |
| 5 | Prior Experience | PE1 | | | | | 0.841 | | |
| | | PE3 | | | | | 0.882 | | |
| | | PE5 | | | | | 0.934 | | |
| | | PE6 | | | | | 0.871 | | |
| | | PE7 | | | | | 0.572 | | |

**Table 2.** *Cont.*

| | | | 1 | 2 | 3 | 4 | 5 | 6 | 7 |
|---|---|---|---|---|---|---|---|---|---|
| 6 | Customer's Attitude | CA1 | | | | | | 0.881 | |
| | | CA2 | | | | | | 0.841 | |
| | | CA3 | | | | | | 0.942 | |
| | | CA4 | | | | | | 0.742 | |
| 7 | Intention to click | ITC1 | | | | | | | 0.912 |
| | | ITC2 | | | | | | | 0.907 |
| | | ITC3 | | | | | | | 0.982 |

Extraction Method: Principal Component Analysis. A Rotation converged in 8 iterations.

### 3.1.2. Reliability Analysis

A reliability test is a psychometric feature that measures particular answers within particular conditions [65]. Cronbach's alpha suggests an internal consistency of scale [61]. According to [66] composite reliability is a more accurate estimate than Cronbach's Alpha. Cronbach's alpha ($\alpha$) threshold is 0.70 and composite reliability (CR) is 0.80 [67]. In the current research, Cronbach's alpha ($\alpha$) suggests that there were only seven variables more than 0.70 (i.e., 0.621–0.927), and composite reliability (CR) was only seven variables more than 0.80 (0.801–0.971). The convergent validity (EVA) indicates the agreement level between the latent construct and its particular measuring tool [60]. The average variance extracted defines the shared average variance between a variable and its measures [68]. Regarding AVE analysis, Table 3 indicates that only seven variables are more than 0.50 [69]. The AVE for the seven variables rated are 0.697–0.932, which shows that the scale items are related. When applying the criteria that specify the rule of Cronbach's alpha higher than 0.70, composite reliability more than 0.80, and average variance extracted more than 0.50.

The factor loading for the 27 items of the 7 variables factors exceeded the threshold of 0.50 [70]. The Kaiser–Meyer–Olkine (KMO) shows data coherence, homogeneity, and interrelationships, and the minimum accepted value is 0.60 [71–73]. Table 3 shows that KMO = 0.887, $p < 0.05 = 0.002$, with Bartlett test for Sphericity ($\chi^2 = 178.958$, $p < 0.05 = 0.001$). The results show that factor analysis is suitable, and all remaining measures are suitable and valid for the Path Analysis. Moreover, Convergent and Discriminant Validity are conducted.

**Table 3.** Internal Consistency and Convergent Validity.

| Credibility | FL | (AVE) | (CR) | ($\alpha$) | Mean | Standard Deviation |
|---|---|---|---|---|---|---|
| C1 | 0.741 | | | | | |
| C2 | 0.795 | 0.821 | 0.841 | 0.802 | 2.866 | 0.661 |
| C3 | 0.814 | | | | | |
| C4 | 0.921 | | | | | |
| **Relevance** | **FL** | **(AVE)** | **(CR)** | **($\alpha$)** | **Mean** | **Standard deviation** |
| R1 | 0.674 | | | | | |
| R2 | 0.774 | 0.774 | 0.801 | 0.798 | 3.049 | 0.582 |
| R3 | 0.769 | | | | | |
| R4 | 0.851 | | | | | |

**Table 3.** *Cont.*

| Originality | FL | (AVE) | (CR) | (α) | Mean | Standard deviation |
|---|---|---|---|---|---|---|
| O1 | 0.877 | | | | | |
| O2 | 0.881 | 0.782 | 0.820 | 0.798 | 3.193 | 0.926 |
| O3 | 0.863 | | | | | |
| O4 | 0.901 | | | | | |
| O5 | 0.931 | | | | | |
| **Avoidance** | **FL** | **(AVE)** | **(CR)** | **(α)** | **Mean** | **Standard deviation** |
| A1 | 0.578 | | | | | |
| A2 | 0.914 | 0.776 | 0.869 | 0.847 | 3.912 | 0.725 |
| A3 | 0.796 | | | | | |
| A4 | 0.687 | | | | | |
| **Prior Experience** | **FL** | **(AVE)** | **(CR)** | **(α)** | **Mean** | **Standard deviation** |
| PE1 | 0.882 | | | | | |
| PE2 | 0.934 | 0.697 | 0.809 | 0.714 | 3.071 | 0.395 |
| PE3 | 0.871 | | | | | |
| **Consumer's Attitudes toward** | **FL** | **(AVE)** | **(CR)** | **(α)** | **Mean** | **Standard deviation** |
| CA1 | 0.881 | | | | | |
| CA2 | 0.841 | 0.887 | 0.932 | 0.907 | 2.714 | 0.807 |
| CA3 | 0.942 | | | | | |
| CA4 | 0.742 | | | | | |
| **Intention to Click** | **FL** | **(AVE)** | **(CR)** | **(α)** | **Mean** | **Standard deviation** |
| ITC1 | 0.912 | | | | | |
| ITC2 | 0.907 | 0.932 | 0.971 | 0.927 | 2.885 | 0.905 |
| ITC3 | 0.982 | | | | | |
| **Kaiser–Meyer–Olkine = 0.887; Sig = 0.002; Bartlett = 178.958; Sig = 0.001** | | | | | | |

### 3.1.3. Validation of Model of Discriminant Validity

Table 3 indicates that all measures are valid, and the model is fit to be used. Discriminant validity shows items' relation with variables [74]. It compares and contrasts the construct coefficients correlation and the AVE square roots, which have to be more than that of other constructs [75], thus, the results are generated from the research model and disseminated. Results of Table 4 indicate that the construct coefficients correlation of all variables is lower than the square roots of (AVE). This indicates that item loadings within its latent construct are more than that for other constructs. Therefore, the study model indicates that discriminant validity is satisfactory, and that generated results can be disseminated.

**Table 4.** Discriminant Validity.

| | R | O | PE | C | CA | ITC | A |
|---|---|---|---|---|---|---|---|
| R | **0.879** | | | | | | |
| O | 0.246 | **0.884** | | | | | |
| PE | 0.064 | 0.076 | **0.834** | | | | |
| C | 0.182 | 0.284 | 0.067 | **0.906** | | | |
| CA | 0.135 | 0.216 | 0.063 | 0.117 | **0.941** | | |
| ITC | 0.190 | 0.393 | 0.053 | 0.178 | 0.122 | **0.965** | |
| A | −0.041 | −0.055 | −0.033 | −0.164 | −0.332 | −0.306 | **0.880** |

### 3.2. Testing Hypotheses

Path Analysis has been used to examine the hypothesis, specifically structural equation modeling, which is usually used to analyze complex causal models [60]. Path Analysis shows four different values, and how much variance of independent variables explains the dependent variable ($R^2$) [70]. Results are shown in Table 5 and Figure 2 as follows:

**Table 5.** The Results of Testing Hypotheses.

| Hypotheses | Path Coefficients (β) | Z-Value | $f^2$ | *p*-Value | $R^2$ | Decision |
|---|---|---|---|---|---|---|
| $PE \rightarrow A$ | 0.085 | 1.220 | - - - - - - - - | $p > 0.05$ =0.222 | | Rejected |
| $O \rightarrow A$ | −0.270 | −3.866 | −0.208 Moderate | $p < 0.05$ =0.000 | **0.129** | Accepted |
| $R \rightarrow A$ | −0.221 | −3.156 | −0.270 Moderate | $p < 0.05$ =0.002 | | Accepted |
| $PE \rightarrow CA$ | 0.103 | 1.636 | - - - - - - - - | $p > 0.05$ =0.102 | | Rejected |
| $O \rightarrow CA$ | 0.222 | 3.520 | 0.171 Moderate | $p < 0.05$ =0.000 | **0.394** | Accepted |
| $R \rightarrow CA$ | 0.250 | 3.971 | 0.306 Moderate | $p < 0.05$ =0.000 | | Accepted |
| $C \rightarrow CA$ | 0.414 | 6.567 | 0.447 Substantial | $p < 0.05$ =0.000 | | Accepted |
| $CA \rightarrow ITC$ | 0.727 | 14.132 | 0.861 Substantial | $p < 0.05$ =0.000 | **0.529** | Accepted |

$R^2$: 0.51 = Strong; 0.33 = Moderate; 0.20 = Weak. Effect size ($f^2$): 0.02 = Small effect; 0.15 = Moderate effect; 0.35 = Substantial effect.

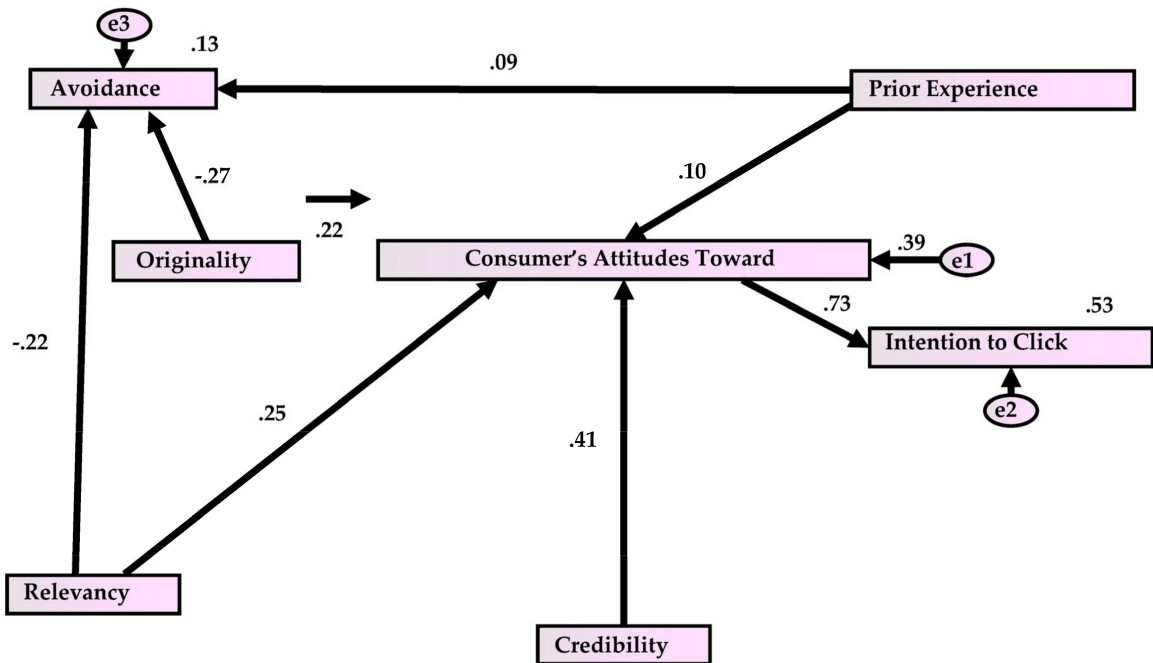

**Figure 2.** The study's model testing.

Analysis results of dimensions (Google ads relevance, originality, and prior experience) hypothesized to affect avoidance:

**H2.** *Display Google ads relevance has a significant negative influence on Display Google ads avoidance. The current hypothesis is accepted, where (β = −0.221; p < 0.05; =0.002). According to the f² value (−0.270), the effect is moderate. This means that an increased display of Google ads relevance leads to a decreased display of Google ads avoidance.*

**H4.** *Display Google ads originality has a significant negative influence on Display Google ads avoidance. The current hypothesis is accepted ($\beta = -0.270$; $p < 0.05$; =0.000). According to the $f^2$ value ($-0.208$), the effect is moderate. This means that an increased display of Google ads' originality leads to a decreased display of Google ads avoidance.*

**H7.** *Prior experience with Display Google ads has a significant negative influence on Display Google ads avoidance. The current hypothesis is rejected ($\beta = 0.085$; $p < 0.05$; =0.222). Results show that prior experience has an insignificant influence on Display Google ads avoidance.*

The variance in display Google ads avoidance that was explained by two independent variables (display Google ads originality and display Google ads relevance) was weak ($R^2 = 0.129$). This value is considered weak and unreliable.

Analysis results of dimensions (Google ads relevance, originality, credibility, and prior experience) hypothesized to affect consumer attitude towards:

**H1.** *Display Google ads relevance has a significant positive influence on consumer attitude towards Display Google ads. The current hypothesis is accepted ($\beta = 0.250$; $p < 0.05$; =0.000). According to the $f^2$ value (0.306), the effect is moderate. This means that increased display Google ads relevance leads to an increase in consumer attitude towards display Google ads.*

**H3.** *Display Google ads originality has a significant positive influence on consumer attitude towards Display Google ads. The current hypothesis is accepted ($\beta = 0.222$; $p < 0.05$; =0.000). According to the $f^2$ value, (0.171), the effect is moderate. This means that increased display of Google ads' originality leads to an increase in consumer attitude towards the display of Google ads.*

**H5.** *Display Google ads credibility has a significant positive influence on consumer attitude towards Display Google ads. The current hypothesis is accepted ($\beta = 0.414$; $p < 0.05$; =0.000). According to the $f^2$ value, (0.447), the effect is substantial. This means that increased display Google ads credibility leads to an increase in consumer attitude towards display Google ads.*

**H6.** *Prior experience with Display Google ads has a significant positive influence on consumer attitude towards Display Google ads. The current hypothesis is rejected ($\beta = 0.103$; $p < 0.05$; =0.102).*

The variance in consumer attitude towards display Google ads that were explained by the above three independent variables (display Google ads originality, display Google ads relevance, and display Google ads credibility) was moderate ($R^2 = 0.394$). This value is good and reliable.

Analysis results of consumer attitude towards Display Google ads effect on the intention to click on Display Google ads.

**H8.** *Consumer attitude towards Display Google ads has a significant positive influence on intention to click on Display Google ads. The current hypothesis is accepted ($\beta = 0.727$; $p < 0.05$; =0.000). According to the $f^2$ value, (0.861), the effect is substantial. This means that increased consumer attitude towards display Google ads lead to an increased intention to click on display Google ads.*

The variance in intention to click on display Google ads that were explained by consumer attitude towards display Google ads was strong ($R^2 = 0.529$). This value points to high reliability.

## 4. Discussion

This research aims to identify and understand the factors that affect the consumer's attitude towards Display Google ads by perceiving the intentions to click. The current research model consists of four independent variables: Display Google ads' credibility, originality, prior experience, relevance, and two dependent variables; intention to click on Display Google ads and Display Google ads avoidance, in addition to one mediator: consumer's attitude towards Display Google ads. The following section includes hypotheses results discussion.

Display Google ads Avoidance: The first set of hypotheses that analyzes the relationships between Display Google ads avoidance and how it is affected by three different

variables: Display Google ads' prior experience, originality, and relevance, resulting in a weak variance ($R^2 = 0.129$). This result may be due to the misleading terms since avoidance is a broad term and could be understood differently by each individual.

Google Display Ads and Prior Experience: Display Google ads' prior experience has a significant negative impact on Display Google ads avoidance. However, the result of the data analysis was inconsistent with this hypothesis ($\beta = 0.085$; $p < 0.05$; =0.222). Past studies examined the impact of prior experience on online ad avoidance, which was accordant with the developed hypothesis [42,56]. A negative experience is more likely to increase the avoidance of online advertisements that were measured by several factors, such as dissatisfaction, lack of incentive, and lack of usefulness [42]. Positive prior experience could increase the impact on SSA credibility and attention to SSA [38]. Consumers that are experienced with advertisements tend to be less tempted to avoid ads [76]. The negative prior experience would affect the user's perceived information processing and the manner of the intended message [56]. Users who are over-experienced with the same advertising clutter might decrease their tendency to avoid internet advertisements [77]. Prior experience is important as it depends on who sends and who receives the message [43], as the information that the receiver of the message has can perform a vital role in the acceptance of the message [44]. Customers' prior experience and trust affect customers' attitudes and behavior [18]. The hypothesis was rejected due to having limited studies that analyzed the correlation between prior experience and ads avoidance within Display Google advertising.

Display Google ads Originality: Display Google ads originality has a significant negative effect on Display Google ads avoidance. The data analysis showed a compatible result with this hypothesis ($\beta = -0.270$; $p < 0.05$; =0.000). Another research further investigated this by stating that originality in terms of creativity can be the element that distinguishes online advertising from traditional advertisements, thus originality is seen as the tool to eliminate the consumers' patterns of avoidance [33]. An original ad is not enough to reduce avoidance, the advertisement must combine originality with other elements, such as usefulness [78], due to the difference between both contexts in which this study investigated advertisements in the context of video advertisement only. The keyword focus on originality, relevance, information, and content, which affect customers' behavior [14]. Massage fit, frame, and focus affect customers' attention and behavior [17]. Real-time Google Ads create new ad data [37]. Original and relevant information increase trust [18].

Display Google ads Relevance: The last hypothesis states that Display Google ads Relevance have a significant negative impact on Display Google ads avoidance. The result of the data analysis was consistent with this hypothesis. Increasing relevant content may reduce ad avoidance due to a lack of interactivity [78]. Users may feel that ads may not be relevant to their interests, which increases the tendency to avoid ads [42]. Consumers might feel that high exposure to information about a relevant ad to them would lead consumers to be less interested in the ad [57]. Advertisers select keywords related to their ad content [11]. The success of online marketing depends on keywords that focus on relevance [14]. The most relevant information performs the main role to trust the advertising platform of Google Ads search engine marketing [18]. The hypothesis was accepted as customers are less likely to avoid ads if the information may be more relevant to their interests.

Conclusions about the Research Hypotheses; Consumer's attitude towards Display Google ads.

The second part of the hypotheses analyzed the relationships between the consumer's attitudes towards Display google ads and how this variable is affected by four other different variables: Display Google ads' prior experience, originality, relevance, and credibility. After the data analysis, results showed that this set of hypotheses gathered a moderate variance ($R^2 = 0.394$).

Display Google ads Prior Experience: Prior experience with Display Google ads has a significant positive effect on consumer attitude towards Display Google ads. The data analysis showed that the results for this hypothesis are inconsistent ($\beta = 0.103$,

$p > 0.05$; 0.002). The research explained that the consumers' perceived attitudes could be shaped by the user's prior experience with the advertisements [22]. Prior experience is important as it depends on the message receivers and senders of the message [43] as the information that the receiver of the message has can perform a crucial role in the acceptance of the message [44]. Customers' prior experience and trust affect customers' attitudes and behavior [18]. The hypothesis was rejected due to having limited studies that analyzed the correlation between prior experience and ads avoidance within Display Google advertising. However, limited studies analyzed the negative correlation between prior experience and consumer attitude. This research found that the consumer has a restricted knowledge of what Display Google ads represent, and for this reason, the hypothesis was rejected.

Display Google ads Originality: Display Google ads originality has a significant positive influence on consumer attitude towards Display Google ads, the current hypothesis was accepted in the data analysis ($\beta = 0.222$; $p < 0.05$; =0.000). Out-of-the-ordinary advertisements can initiate and stimulate consumer attitudes and purchasing patterns [79]. The ads' originality influences consumers' attitudes positively [55]. Original and creative advertisements can help consumers to expedite recalling the advertisements easily [80]. Social media ads increase consumer liking, awareness, and action, which have a strong correlation with consumers' attitudes [13]. The keyword focus on originality, relevance, information, and content, which affect customers' behavior [14]. Massage fit, frame, and focus affect customers' attention and behavior [17]. Real-time Google Ads create new ad data [37]. Original and relevant information increase trust [18]. The acceptance of the hypothesis results from the concept in consumers tend to have more positive attitudes toward original advertisements that are remarkable and unprecedented.

Display Google ads Relevance: The result for the hypotheses developed for this variable towards consumer attitude was consistent as previewed in the data analysis ($\beta = 0.250$; $p < 0.05 = 0.000$). In the context of banner ads, a consumer would rather act positively toward an ad if the content were more relevant to them [57]. Relevance is an important variable to influence the consumer's attitude (Al-Khasawneh, 2009). When more ads are perceived to be relevant this will reflect positively on the consumer's attitudes [57]. Advertisers select keywords related to their ad content [11]. Focusing on relevance is very important for successful online marketing [14]. The most relevant information performs the main role to trust the advertising platform of Google Ads search engine marketing [18].

Display Google ads Credibility: The results were consistent with the hypotheses that were developed in the previous literature ($\beta = 0.414$; $p < 0.05 = 0.000$). Most of the studies in the literature were supportive of this result [24,81]. Credibility performs a vital role in the customers' insight towards the content of ads, which improves the positive consumer attitude [82]. The trustworthiness and attractiveness of an ad can highly impact a consumer's attention [40], for example, a consumer's first impression of an ad can extremely influence its credibility of it, thus making it an important variable to be examined [39]. Positive thinking related to the ad's credibility enhances the attitudes of the customer toward the ad [41]. Google ads create a positive image of products and/or services, and increase customer engagement and sales [12]. Trust in social media tools affects consumer attitudes [13], brand awareness, and customers' buying decisions [15]. Confidence and personal experience are important for customers' trust [18]. Finally, the hypothesis was developed due to the importance of the credibility of the ad's context in customers' minds will result in more positive intention toward the Display Google ads.

*Conclusions about the Research Hypotheses; Intention to Click*

The relation was investigated between the consumer's attitude towards Display Google ads and its impact on the intention to click, which gives rise to a strong variance ($R^2 = 0.529$); however, the hypothesis was accepted ($\beta = 0.727$; $p < 0.05$; =0.000). Most of the variables demonstrated in the model had a direct effect on consumer attitude towards Display Google ads resulting in an intention to click. A previous study declared this by implying that the more the advertisement was convenient to the consumer the more

they will be motivated to perceive the ad and click on it [29]. Moving into the context of sponsored search advertising, a study concluded that consumers' attitudes could be considered a vital measurement in examining the intention to click [22]. The consistency of the hypothesis was because consumer attitude is considered a key factor when determining the effectiveness and the efficiency of an ad resulting in the intention to click. Positive prior experience could increase the impact on SSA credibility and attention to SSA [38]. A negative prior experience enhances the likelihood of an ad being avoided [42]. Prior experience is important as it depends on both the message sender and the receiver [43]. Advertisers select keywords related to their ad content [11]. Information is vital for message acceptance [44]. Credibility is about the content of ads, which affects consumer attitude [82]. Ad credibility enhances persuasion and convincing [39]. Ad trust and attraction affect consumers' attention [40]. The ad's credibility contributes to customers' attitudes toward the ad [41]. Social media ads have a strong correlation with consumers' attitudes [13]. Google ads' image affects customer engagement and sales [12]. Trust in social media tools affects consumer attitudes [13], and customers' buying decisions [15]. Ads' originality, relevance, information, and content affect customers' behavior [14]. Massage fit, frame, and focus affect customers' attention and behavior [17]. Customers' prior experience and trust affect customers' attitudes and behavior [18]. Customers' prior experience, originality, and relevant information increase confidence and trust, which affects customers' attitudes and behavior [18].

## 5. Conclusions

In conclusion, this research paper focuses on acknowledging the impact of variables that perform an important role in affecting the consumers' attitude towards Display Google ads, resulting in consumers either avoiding or clicking. Marketers need to understand the several factors that influence their online advertisement, which leads an ad to be effective or not. Even though many previous studies examined the factors influencing the consumer's attitude or behavior toward online advertising, no other research combined factors that affected two independent variables: avoidance and intention to click. Thus, this research aims to fill the gap in the existing literature. To enhance our understanding of such independent variables, an extended model has been developed to enrich our knowledge concerning the consumers' attitude towards Display Google ads. The study aims to provide a complete image of the variables that encourage users to click on Display Google ads. The constructed model is made up of four dependent variables (Display Google Ads prior experience, originality, relevance, and credibility), two independent variables (Display Google Ads avoidance and intention to click), and one mediator (consumers attitude). The current research utilized a qualitative data collection method in which an online survey was used to distribute questionnaires to the targeted population. Then, Structural Equation Modeling (SEM) was used to analyze data, which evaluates the relationship between the developed variables; this was complete through the implementation of a software program (AMOS 22). The current research's findings conclude that the first set of variables influence Display Google ads avoidance, two (Display Google ads Originality and Relevance) have a significant impact, whereas one variable (Display Google ads Prior Experience) does not influence Display Google ads Avoidance. However, concerning the second set of variables affecting the consumers' attitude, which results in the intention to click, three (Display Google ads originality, relevance, and credibility) have a significant impact, yet one variable (Display Google ads prior experience) does not have an impact on consumers' attitude. Finally, the outcomes of this research show both practical and theoretical implications that can be employed by marketing and researchers in the future.

### 5.1. Future Research

The current study has many limitations. First, although the online survey was answered by a significant sample of the whole population, still the majority of the respondents were millennials. Hence, a future study can examine the effect of consumers' attitudes

towards Display Google ads on a larger scale of age groups to ensure that the survey is correctly distributed throughout the targeted population. Furthermore, this current research was tested and analyzed on Display ads using Google, as the main platform. However, future research can be extended to other online advertising platforms, such as Yahoo or Bing. The study in hand used a quantitative approach to gather primary information, which might lack a deep understanding of the consumer's actual feelings and behaviors. Therefore, future research can be adapted to use a qualitative approach to create a bigger image of the correlation between consumers' attitudes and the intention to click.

### 5.2. Theoretical Implications

The current research model was developed to investigate and analyze the dominant factors related to consumers' attitudes towards Display Google ads, which may influence their intention to click. Although several studies examined consumers' intention to click on mobile advertising, sponsored search advertising, and internet advertising, the concept of Display Google ads has not been thoroughly acknowledged. Thus, the objective of the current research is to fill the gap reviewed in the previous literature. After a thorough revision of the previous studies, the established model of the current study adds to the correlation of various variables with the tendency to act toward ads, hence resulting in consumers clicking or avoiding ads. The development of the current research model was based on an extension of existing models, which examined the effect of sponsored search advertising on the consumer's intention to click on them [22]. Yet, the current research is set to be the first to combine a theoretical model that includes diverse factors that influence the consumers' attitude in the Display Google ads context, which leads consumers to click or avoid ads. Furthermore, the theoretical model of this research has not been examined in the Middle East before. For such reasons, researchers and marketers specifically would utilize this model in the future. Moreover, the current research demonstrates the factors that impact the tendency to click or avoid ads, which will help marketers in the process of creating content with the aim of it being recognized by online users. The model illustrates new relationships, such as the effect of credibility on consumers' attitudes, which resulted to be positive. Furthermore, the prior experience was studied to understand the consumers perceived background information influence on their attitude. Finally, the model content contained two characteristics in the context of creativity, originality and relevance, improving the psychological understanding of the consumer's attitude towards Display Google ads.

### 5.3. Practical Implications

The theoretical model of this research found three out of four variables, which are Display Google ads' credibility, originality, and relevance, have a significant positive effect on customers' attitudes toward Display Google ads, which may lead to an influence on the intention to click. However, Display Google ads' credibility showed the highest significant impact on consumers' attitudes toward Display Google ads. Moreover, consumers' attitude towards Display Google ads is the main influence on the consumers' likelihood to click; however, it should be noted that this occurred due to the inflation caused by the impact of all variables discussed above. Customers' attitudes toward online ads have a significant influence on the customers' likelihood to click, as internet advertising may create more positive attitudes. Organizations' managers have to focus on ads' relevance, originality, and credibility because they directly affect customers' attitudes toward Display Google ads. Moreover, marketing managers have to increase awareness campaigns to affect the attitude toward their brands positively. Finally, managers have to use different social media tools to affect customers' attitudes and intentions to click.

Originality and relevance, in the context of creativity, have a moderate influence on Display Google ads avoidance in the research's developed theoretical model, which can help researchers or marketers to understand the negative psychological influence that causes online users to skip or avoid ads. Researchers or marketers would highly

benefit from this newly developed model to enhance online users' positive attitude towards internet advertising and decrease the likelihood of ads being avoided and skipped.

As for the Display Google ads prior experience, the developed hypotheses were rejected, which showed the variables have neither an impact on Display Google ads avoidance nor the consumers' attitude towards Display Google ads. The consumer seems to be uninterested in ads that seem known to them, which leads to meaningless information being added to their knowledge. Hence, as a third-world country, the technology has not been completely understood by the whole population, which results in a lack of prior knowledge of different online ad categories. Correspondingly, marketers should intensify online consumers' knowledge about the different categories of online advertisements.

**Author Contributions:** Conceptualization, M.A.K. and S.A.-H.; methodology, A.-A.A.S.; software, A.-A.A.S.; validation, M.A.K. and S.A.-H.; formal analysis, A.-A.A.S.; investigation, R.A.-D., S.H. and S.S.; resources, R.A.-D., S.H. and S.S.; data curation, R.A.-D., S.H. and S.S.; writing—original draft preparation, R.A.-D., S.H. and S.S.; writing—review and editing, A.-A.A.S.; visualization, A.-A.A.S.; supervision, M.A.K. and S.A.-H.; project administration, A.-A.A.S.; funding acquisition, M.A.K. and S.A.-H. All authors have read and agreed to the published version of the manuscript.

**Funding:** This research received no external funding.

**Data Availability Statement:** Primary data is available on request.

**Acknowledgments:** Thanks to Middle East University for its continuous research support.

**Conflicts of Interest:** The authors declare no conflict of interest.

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
