# Peer review of "Consumer’s Attitude towards Display Google Ads"

_futureinternet, doi:10.3390/fi15040145_

Round 1

Reviewer 1 Report

Thanks for giving me the chance to read this very interesting article.

This study, after taking into consideration previous literature, develops a theoretical model combining four 24 variables (Display Google ads' Prior Experience, Originality, Relevance, and Credibility) that lead 25 to affecting Display Google ads' Avoidance and Intention to Click, with one mediator (Consumer’s 26 Attitude). A quantitative methodology has been employed, in which an online survey has been used 27 to collect data, which were analyzed by AMOS. The data analysis results show that three independent variables positively impact the intention to click, however, credibility has the highest value. While Display Google ads prior experience had no impact on the intention to click.

The article is methodologically sound and results are without any doubt interesting.

In this sense it contributes to the existing literature about Google Ads.

However, in its current form the article suffers from some major flaws that should be fixed before publication.

1) STRUCTURE

The Introduction mixed theoretical background, literature review and research questions

I suggest to create at least different subheading for Introduction, in which you should introduce the paper, Theoretical Background/Literature review and Research questions

2) THEORETICAL BACKGROUND

The paper can be really interesting for all those scholars interested in social media (media studies, marketing, media literacy, etc.) but in its current state it seems it is only specific for those studying Google ads. And that is a petty.

The author(s) concisely mention this in this sentence “Due to the fast development of technologies and the internet, the number of online 38 users is increasing dramatically [1]. At the same time, social media tools are increasingly 39 leading to changing marketing strategies and approaches [2]. Nowadays, social media 40 tools are very important for all organizations, because they help to improve the relation-41 ship with customers [3] by providing customers with updated information and services 42 [1,4], and enhancing customers’ engagement [5].”

I suggest to give more space to these reflections (this journal has no word limitation) to increase the paper readability and citability.

In particular:

A) The number of online users is increasing

Here I would add some data about internet users

Most importantly it is seminal to at least mention young people and smartphones.

See for instance:

Tejedor, S.; Cervi, L.; Pérez-Escoda, A.; Tusa, F. Smartphone usage among students during COVID-19 pandemic in Spain, Italy

and Ecuador. In Eighth International Conference on Technological Ecosystems for Enhancing Multiculturality, Salamanca, Spain, 21–23

October 2020; ACM: New York, NY, USA, 2020; pp. 571–576.

B) Social media and marketing

Same. There is a wide literature about social media and marketing that can help you enriching the discussion.

See for instance:

Li, F., Larimo, J., & Leonidou, L. C. (2021). Social media marketing strategy: definition, conceptualization, taxonomy, validation, and future agenda. Journal of the Academy of Marketing Science, 49, 51-70.

C) And most importantly.

Social media and marketing are crucial issues involving digital and media literacy, since the users use and ability to navigate the internet depends on their digital skills.

Although this is not the focus of your paper this topic must be acknowledged.

This reference can help:

Cervi, L.; Tornero, J.M.P. Changing the Policy Paradigm for the Promotion of Digital and Media Literacy. The European Challenge.

In Pursuing Digital Literacy in Compulsory Education: Reconstructing the School to provide Digital Literacy for All; Peter Lang Inc.: New

York, NY, USA, 2011; pp. 50–70.

3) LANGUAGE

I suggest the author(s) to use a native editor/proofreader since the language in certain sections can be hard to follow.

Good Luck!

Author Response

Dear respected Prof.

I responded to all your fruitful comments one by one. Please check.

Mant thanks for your fruitful comments.

Best regards.

Reviewer 2 Report

The manuscript addresses consumers' attitudes towards Display Google Ads. Altought a complex statistical analysis is conducted, I cannot se an innovative aspect of the study. The research topic is not well substantiated, the advances compared to existing research is poorly described. Moreover, the concept of the research methodology is vague. Why did the authors employ exploratory factor analysis if they state hypotheses? In this case research questions would be appropriate. Otherwise, confirmatory factor analysis would be the first step. For these reasons, unfortunately, I do not recommend the paper for acceptance.

Author Response

Dear respected prof.

May I request you clarify what exactly is needed? I could not understand what should I do. I tried to improve it according to other reviewers' comments.

Best regards

Reviewer 3 Report

The paper covers an interesting topic and offers an interesting empirical research into the topic studied.  

The paper presents an adequate understanding of the current literature in the field in a way which is useful to the development of our understanding in the area it addresses

By addressing these themes, this research makes a useful contribution

The paper does explain and justify its research design and the collection of data. Hypotheses of the paper are clearly stated and adequately tie together the other elements of the research (such as literature, data and critical perspectives)

Author Response

Dear respected Prof.

Thank you for your comments and best regards.

Reviewer 4 Report

Manuscript ID: futureinternet-2247476

The manuscript deals with a potentially interesting topic. In its current form the text needs some major changes and clarifications for the study’s potential contribution to be shown. In particular, the authors might wish to

1)     Provide a clear presentation of the analytical steps taken. The review part that results into proposing a theoretical analysis models needs to be written in a tighter and concise manner inclusive of what is new as proposed here. In doing that the schematic presentation of the analysis also needs to be clarified. The focus is on intention to click yet avoidance which is another decision is also analyzed? In short figure 1 is not clear (also includes arrows that are not tested?) please clarify in view of illustrating the potential contribution of the proposed schema of relations.

2)     Provide an in depth discussion of how the suggested hypotheses offer new insights to the issue under study. What is new about Google Ads as a case study and how these insights might enrich our understanding of the ‘click’ process?

3)     In the methodology part more information is needed, e.g. please report area of respondents, report items per variable, descriptive statistics, etc.

4)     Overall, the text needs to be rewritten in order to provide a tight, concise and informative text that is clear to read and follow.

5)     I would strongly suggest that the authors use an English speaking editor to improve the study in terms of syntax and presentation of arguments, structure etc. As is the text is unclear and difficult to follow. 

Author Response

Dear respected Prof.

I responded to your comments as much as I can. Most of the comments are unclear and do not clearly indicate what I should do.

Thank you and best regards.

Round 2

Reviewer 1 Report

Publishable

Author Response

Dear Sir,

Kindly, find attached the final version.

Best regards

Reviewer 2 Report

Figure 1 is not sufficiently placed. Please, correct formatting.

Figure 2 as well.

I recommend correcting - one sentence is not a single paragraph - see line 579. Please, merge with other text.

The authors do not provide sufficient content under the subchapter "Practical implications". The authors should provide at least three practical implicatons.

Table 2: Why the component matrix is not cleaned? Please, provide cleaned component matrix. A7 - 2, factor loading .524 is too high. The same applies to PE7 - 3. Lines 303-306 - adding the reference to statistical literature is essential.

Author Response

Dear Sir,

Kindly find attached the final version adjusted as requested.

Best regards

Figure 1 is not sufficiently placed. Please, correct formatting. Done

Figure 2 as well. Done

I recommend correcting - one sentence is not a single paragraph - see line 579. Please, merge with other text. Done

The authors do not provide sufficient content under the subchapter "Practical implications". The authors should provide at least three practical implicatons. I added three points.

Table 2: Why the component matrix is not cleaned? Please, provide cleaned component matrix. A7 - 2, factor loading .524 is too high. The same applies to PE7 - 3. Lines 303-306 - adding the reference to statistical literature is essential. Adjusted accordingly.

Best regards

Reviewer 4 Report

The manuscript has been improved. 

Author Response

Dear Sir,

Kindly, find attached the final version.

Best regards.
